# Accelerating the Finite-Element Method for Reaction-Diffusion Simulations on GPUs with CUDA

**DOI:** 10.3390/mi11090881

**Published:** 2020-09-22

**Authors:** Hedi Sellami, Leo Cazenille, Teruo Fujii, Masami Hagiya, Nathanael Aubert-Kato, Anthony J. Genot

**Affiliations:** 1Department of Computer Science, The University of Tokyo, Tokyo 113-8654, Japan; hedi@sellami.dev (H.S.); hagiya@is.s.u-tokyo.ac.jp (M.H.); 2Department of Information Sciences, Ochanomizu University, Tokyo 112-8610, Japan; leo.cazenille@gmail.com; 3LIMMS/CNRS-IIS, UMI2820, The University of Tokyo, Tokyo 153-8505, Japan; tfujii@iis.u-tokyo.ac.jp

**Keywords:** Finite-Element Methods, GPU, CUDA, non-linear PDEs, reaction-diffusion

## Abstract

DNA nanotechnology offers a fine control over biochemistry by programming chemical reactions in DNA templates. Coupled to microfluidics, it has enabled DNA-based reaction-diffusion microsystems with advanced spatio-temporal dynamics such as traveling waves. The Finite Element Method (FEM) is a standard tool to simulate the physics of such systems where boundary conditions play a crucial role. However, a fine discretization in time and space is required for complex geometries (like sharp corners) and highly nonlinear chemistry. Graphical Processing Units (GPUs) are increasingly used to speed up scientific computing, but their application to accelerate simulations of reaction-diffusion in DNA nanotechnology has been little investigated. Here we study reaction-diffusion equations (a DNA-based predator-prey system) in a tortuous geometry (a maze), which was shown experimentally to generate subtle geometric effects. We solve the partial differential equations on a GPU, demonstrating a speedup of ∼100 over the same resolution on a 20 cores CPU.

## 1. Introduction

In the past two decades, the architecture of Graphical Processing Units (GPUs) have made them the tool of choice in scientific computing to solve massively parallel problems [1,2]. CPUs spend a sizable fraction of their transistors budget on caching and control units. This allows CPUs to quickly serve data that are often accessed (e.g., for database server) and to handle complex and varied flows of instructions (e.g., out-of-order or speculative execution), but this comes at a cost of a reduced number of computing units (arithmetic logic units). By contrast, GPUs spend almost all their transistor budget on arithmetic logic units, because they were initially designed to handle homogeneous but massively parallel flows of instructions (typically the rendering of graphical scenes, which heavily relies on linear algebra). This focus on computing units enables GPUs to beat CPUs on problems that can be formulated in a parallel manner. For instance, GPUs typically accelerate dense matrix multiplication (a classical, though somewhat artificial, benchmark for supercomputers) by a factor of 6 over CPUs [3], and most supercomputers now embark GPUs to accelerate their computations.

This speedup comes at a cost: it is necessary to write specific code to fully exploit the parallelism of GPUs. This porting has been facilitated by CUDA libraries [4], which provide a middle-level programming interface for NVIDIA GPUs and have enabled GPU-enabled software in many areas of scientific computing. For instance, the meteoric rise of deep learning in the past decade is often attributed to the availability of GPU-enabled frameworks (TensorFlow [5], PyTorch [6], …). Molecular dynamics has also tremendously benefited from GPUs [7,8,9], and most major packages are now GPU-accelerated (LAMMPS, AMBER, CHARMM, …). In many of these cases, GPUs offer sizable speed up on real-world problems (typically ∼10–100×).

GPUs are also a powerful tool for the Finite Element Method (FEM) [10]. These simulations are routinely used by scientists and engineers to solve Partial Differential Equations that describe physical phenomena on prescribed geometries: mechanical stress, heat dissipation, dispersion of chemicals, and so on. The literature on FEM and GPUs is now abundant [11,12,13,14,15,16,17,18,19,20,21,22], yet FEM software still heavily rely on CPUs for their computations, and few support GPUs (e.g., COMSOL [23], a popular commercial software, does not support GPUs). In absence of routine GPU-acceleration for FEM software, it is difficult to judge if a particular physical PDE could benefit from GPUs without actually implementing the FEM resolution at a low level on the GPU.

The goals of this paper are twofold. First and foremost, we set out to investigate if the community of DNA nanotechnology and microfluidics could benefit from GPU-accelerated simulations. In recent years, progresses in DNA nanotechnology have enabled the programming of chemical reaction networks with advanced dynamics (oscillations, multi-stability …) [24,25,26,27,28,29] and information-processing capabilities [30,31,32,33,34,35,36]. Coupled to microfluidics and microfabrication which allow the massive screening of experimental conditions, the fabrication of chemical reactors with arbitrary geometries or the control of chemical reactions with electrical or opticals signals [37,38,39,40,41,42,43], the community of DNA nanotechnology has been exploring reaction-diffusion as a way of building and programming matter at the micro-scale [44,45,46,47,48]. Simulations of PDE play an essential role to design, prototype and debug these DNA-based systems. But the nonlinear nature of their chemistry and the complex geometries of their reactors make simulations computationally intensive. For instance, microfluidic channels often turn at a right angle, but such geometries with sharp corners are known to cause difficulties for FEM [49]. While there has been a body of works addressing GPUs for reaction-diffusion (Table 1 and [50,51,52,53,54]), it is often limited to rectangular reactors (using the Finite Difference Method to compute diffusion). When arbitrary geometries are considered (using FEM), it is often applied to problems with different physics (e.g., propagation of cardiac waves at the surface of the heart). Overall, only a few authors [51] seem to have addressed the question of interest for DNA nanotechnology: accelerating the simulation of nonlinear biochemical reactions and diffusion in arbitrary geometries.

The second goal of this paper is to revisit the typical speedup of GPUs for FEM with current hardware and software. Much of the literature on GPU-accelerated FEM dates back to ∼2010–2015 (Table 1). The GPU speedup was often measured against a single CPU thread or a few CPU cores, and the power consumption—a metric that is becoming increasingly important—was rarely, if ever, reported. But the performances of CPUs have boomed since then (and their power consumption as well), and it is not uncommon for a desktop computer to embark a CPU with 16 cores or more (and to draw up to ∼300 W). GPUs have also boomed thanks to the coming of age of deep learning, which has prompted massive investment on GPU-based technologies. Their memory and their precision (two of their historical weak points) have increased in the past decade. Professional GPUs now routinely embark ∼10–20 GB of RAM and support double precision computations, which enable much finer grained simulations than was possible a decade ago. On the software side, CUDA libraries have matured in the past decade, with noticeable gains in performance. It is thus an interesting exercise to revisit the gain of performance with state-of-the-art CPUs (20 cores), GPUs (Titan V) and software libraries (CUDA 10).

Reaction-diffusion systems are a perfect example of PDEs with complex dynamics, describing chemicals diffusing freely in a space while reacting with each other. Starting with the Belouzov-Zhabotinsky reaction [58] and Turing patterns [59], such systems are known to create dynamic structures (such as traveling waves and spirals) and steady state patterns. Advances in the field of molecular programming has allowed the programming of reaction-diffusion with chemical reaction networks [24,26,44,45,46,60,61,62,63,64,65,66,67,68]. That approach has opened the door to intricate systems, with the caveat that the simulation of such systems, a necessary step in the design process, becomes increasingly expensive as the systems get more and more complex.

Here we solve reaction-diffusion equations in arbitrary geometries on GPUs with FEM [47,48,61]. This method discretizes in space and time a continuous PDE, formulating it as a problem of linear algebra for which the architecture of GPUs is uniquely appropriate. As a toy model, we study a nonlinear chemical system (a predator prey chemical oscillator) [27,28] in a tortuous geometry (a maze) [39]. This system was studied experimentally in a microfluidic device and captures the essence of how boundary conditions influence dynamics [39]. It generates traveling waves of preys that are closely followed by predators. The waves closely interact with the geometry of the maze: they propagate along the walls, split at junctions, and terminate at cul-de-sacs. Thus numerically solving this system with FEM is a challenging but informative case to study. The fixed geometries and the boundary conditions (no flux through the walls of the reactor) simplifies the FEM framework, while keeping a rich spatio-temporal dynamics. Moreover, the maze presents many sharp corners, which are known to be challenging for FEM, and represents a good testbed for porting the method to GPUs. A workflow of our methodology can be found in Figure 1.

## 2. Materials and Methods

### 2.1. Chemical System

We consider a biochemical predator prey systems as described by [27,28]. It consists of a prey species *N*, which is a DNA strand that replicates enzymatically, and which is predated by a predator strand *P*. This system mimics the dynamics of the Lotka-Voltera oscillator, and produces stable oscillations of concentration for days when run in a closed test tube.

The PDEs describing the dynamics of the system are
(1)$$\partial_{t}N=C_{r}\frac{N}{1+bN}-C_{p}\frac{NP}{1+bP}-\epsilon +\Delta N$$
(2)$$\partial_{t}P=C_{p}\frac{NP}{1+bP}-\epsilon +\Delta P$$
with Neumann boundary conditions (no flux through the wall of the reactor). The first term in the prey equation describe its auto-catalytic growth, the second terms describes its predation by the predator *P*. The laplacian term describe the diffusion of the prey. The first term in the predation equation describes the growth of the predator induced by the predation of the prey. We found that the null state was locally unstable, and that numerical error caused by the resolution would grow exponentially and cause the spontaneous generation of species. To regularize the equations, we added a small artificial term −ϵ, which was chosen small enough so as not to affect the global dynamic of the system. To prevent negative concentrations caused by this term, we set concentrations to 0 when they become negative. The parameters for the simulations we present are: $C_{r}=C_{p}=0.2$, b=0.1 and $\epsilon =10^{-13}$. The initial concentrations for the prey $N_{0}$ and the predator $P_{0}$ are taken to be 1 in the starting area (a small zone at the bottom of the maze) and 0 elsewhere.

### 2.2. Finite Element Method

We briefly present the Finite Element Method applied to our case [10]. We start from the reaction-diffusion PDE with Neumann boundary conditions (no flux through the boundaries ω of the reactor Ω)
(3)$$\begin{array}{ccc} \partial_{t}N & = & f_{1}\left(N,P\right)+\Delta N\\ \partial_{t}P & = & f_{2}\left(N,P\right)+\Delta P \end{array}$$
where N(x,t) and P(x,t) are the concentration of species *N* and *P* at position *x* and time *t*. The function $f_{i}$ encodes the local chemical reactions that produce or remove the species *N* and *P*. It is smooth, typically a polynomial function (in the case of mass action chemical kinetics) or rational function (in the case of enzymatic kinetics) of the concentrations of species.

We spatially discretize the reactor Ω into a mesh. The mesh partitions the region Ω into simple and non-overlapping geometrical cells (typically triangles), whose vertexes are called the nodes (Figure 1). Contrary to the fixed grid of the finite difference method, meshes allow for a flexible attribution of the computational budget. The physics of regions with an irregular geometry (e.g., turns in a maze) can be better fit by devoting more cells to their approximation. It must be noted that the problem of tessellating a geometric region into a mesh is an active topic of research, and will not be covered in this paper (we produce the mesh with Mathematica [69]). A mesh is associated with a basis of function $\varphi_{i}\left(x\right)$, which is commonly defined to be 1 at the node *i*, and null on the other nodes. Again, there are many possible ways of defining how $\varphi_{i}\left(x\right)$ varies between nodes (piece-wise linear, piece-wise quadratic, …), and we use the basis of functions selected by the FEM software. Following the Galerkin method [70], we search for solution of Equation (Section 2.2) on the basis of functions $\varphi_{i}\left(x\right)$
(4)$$\begin{array}{ccc} N(x,t) & = & \sum_{i=1}^{m}N_{i}\left(t\right)\varphi_{i}\left(x\right)\\ P(x,t) & = & \sum_{i=1}^{m}P_{i}\left(t\right)\varphi_{i}\left(x\right) \end{array}$$
where $N_{i}\left(t\right)$ and $P_{i}\left(t\right)$ are the concentrations of species *N* and *P* at the node *i* and time *t*, and *m* is the number of nodes. Additionally we assume that the nodes are close to each other and that $f_{i}$ are sufficiently smooth, so that $f_{i}$ can be linearly interpolated between the nodes
(5)$$\begin{array}{ccc} f_{1}\left(N\left(x,t\right),P\left(x,t\right)\right) & = & \sum_{i=1}^{m}f_{1}\left(N_{i}\left(t\right),P_{i}\left(t\right)\right)\varphi_{i}\left(x\right)\\ f_{2}(N\left(x,t\right),P\left(x,t\right)) & = & \sum_{i=1}^{mm}f_{2}\left(N_{i}\left(t\right),P_{i}\left(t\right)\right)\varphi_{i}\left(x\right) \end{array}$$

We now expand Equation (Section 2.2) on the Galerkin basis
(6)$$\begin{array}{ccc} \sum_{i=1}^{m}\partial_{t}N_{i}\left(t\right)\varphi_{i}\left(x\right) & = & \sum_{i=1}^{m}f_{1}\left(N_{i}\left(t\right),P_{i}\left(t\right)\right)\varphi_{i}\left(x\right)+\sum_{i=1}^{m}N_{i}\left(t\right)\Delta \varphi_{i}\left(x\right)\\ \sum_{i=1}^{m}\partial_{t}P_{i}\left(t\right)\varphi_{i}\left(x\right) & = & \sum_{i=1}^{m}f_{2}\left(N_{i}\left(t\right),P_{i}\left(t\right)\right)\varphi_{i}\left(x\right)+\sum_{i=1}^{m}P_{i}\left(t\right)\Delta \varphi_{i}\left(x\right) \end{array}$$

To derive the matrix differential equation, we define the element damping matrix **D**
$$D_{ij}=\int_{\Omega }\varphi_{i}\varphi_{j}d\Omega$$
and the element stiffness matrix **S**
$$S_{ij}=\int_{\Omega }\nabla \varphi_{i}\cdot \nabla \varphi_{j}d\Omega .$$

The matrices **D** and **S** are sparse and positive-definite. They are mostly filled with 0 because $D_{ij}$ and $S_{ij}$ is 0 when the nodes *i* and *j* are not close to each other. Thanks to Green’s first identity and the Neumann boundary conditions
(7)$$\int_{\Omega }\varphi_{j}\Delta \varphi_{i}d\Omega =\int_{\omega }\varphi_{j}\left(\nabla \varphi_{i}\cdot n\right)d\omega -\int_{\Omega }\nabla \varphi_{j}\cdot \nabla \varphi_{i}d\Omega =-S_{ij}$$
where ω is the boundary of Ω and **n** the normal vector on this boundary. The integral over the boundary ω is null because of the Neumann conditions (no flux). We take the inner product on $L^{2}\left(\Omega \right)$ by multiplying Equation (Section 2.2) by $\varphi_{j}\left(x\right)$ and integrating over Ω. We obtain the following matrix ordinary differential equation
(8)$$\begin{array}{ccc} D\partial_{t}N\left(t\right) & = & Df_{1}\left(N\left(t\right),P\left(t\right)\right)-SN\left(t\right)\\ D\partial_{t}P\left(t\right) & = & Df_{2}\left(N\left(t\right),P\left(t\right)\right)-SP\left(t\right) \end{array}$$
where the function *f* is threaded on the vectors N(t) and P(t).

This differential equation comprises two operators (the chemical reaction operator and the diffusion operator) with different physical properties. We integrate this equation using the split-operator method [71], splitting the reaction and diffusion operators and applying them alternatively to advance in time. More precisely, we discretize in time by defining $N^{k}=N\left(k\tau \right)$ and $P^{k}=P\left(k\tau \right)$, where τ=0.01 is a small time step and the integer *k* is the number of steps taken. At each time-step *k*, we first apply the chemical reaction operator (with Euler explicit method) to the vectors $N^{k}$ and $P^{k}$, yielding intermediate vectors ${\tilde{N}}^{k}$ and ${\tilde{P}}^{k}$
(9)$$\begin{array}{ccc} {\tilde{N}}^{k} & = & N^{k}+\tau f_{1}\left(N^{k},P^{k}\right)\\ {\tilde{P}}^{k} & = & P^{k}+\tau f_{2}\left(N^{k},P^{k}\right) \end{array}$$

We then apply the diffusion operator with a time-step τ to the intermediate vectors ${\tilde{N}}^{k}$ and ${\tilde{P}}^{k}$ to obtain the vectors $N^{k+1}$ and $P^{k+1}$ at the step k+1 by solving the linear system
(10)$$\begin{array}{ccc} \left(D+\tau S\right)N^{k+1} & = & D{\tilde{N}}^{k}\\ \left(D+\tau S\right)P^{k+1} & = & D{\tilde{P}}^{k} \end{array}$$

We chose the implicit Euler method for the diffusion step due to its better stability than the explicit version. We solve this linear equation with the conjugate gradient method (CGM) [72,73] because the matrix (D+τS) is positive definite, and it only requires computing dot products and sparse matrices-vector products operations which are well adapted to GPUs.

By splitting the reaction and diffusion operators, we reduce the resolution of the PDE to operations that are well adapted to the parallel architectures of GPU. The reaction operator is a point-wise operator: it is already vectorized and easy to apply in a GPU. As for the diffusion operator, it reduces to linear algebra on sparse matrices and dense vectors, for which GPUs are highly performing.

### 2.3. Assembly of Stiffness and Damping Matrices

We used Mathematica [69] to assemble the stiffness and damping matrices. We started from a PNG image of a maze, which we binarized and converted into a geometrical region. We then discretized this region into a mesh with the FEM package of Mathematica, and assembled the corresponding damping and stiffness matrices. We controlled the size of the matrices (and the number of elements in the mesh) by changing the maximum allowed cell size during the discretization (smaller cells yield larger matrices). We then exported the matrices in CSV format.

### 2.4. Resolution of the Matrix Differential Equations

The bulk of the resolution was handled at a high level by a python program, which in turns called a C++ library accelerated using CUDA libraries (including CuBLAS [74] and CuSparse [75]) and home-made CUDA kernels to solve equation at a low level on the GPU. After parsing the damping and stiffness matrices from the CSV file, the python program loaded them onto the GPU.

For the diffusion operator, we solve the linear system with the conjugate gradient method (CGM) [72,73,76], as described in Algorithm 1. Each iteration of the CGM consists mostly of three operations: (1) products between a sparse matrix and a vector (which are handled by CuSparse [75]); (2) additions of two vectors (detailed in Algorithm 2); and (3) dot products between vectors. We implemented a version of the dot product that we optimized for GPUs, with increased performances compared to naive dot product algorithms (Algorithms 3 and 4). We iterate the solution until the relative error of the residual decreases below 0.001.
**Algorithm 1** User-level algorithm. Import Damping matrix, Stiffness matrix, Mesh data N← initial prey state P← initial predator state System.State←N,P

System.Reactions←
**define chemical reactions**
 System.DiffusionParameters← Damping matrix, Stiffness matrix, Threshold precision 
**for** 
i←0,T/τ
**do**
        System.ApplyDiffusion(τ)               ▹ Basic conjugate gradient method        System.ApplyReaction(τ)         ▹ Apply whatever function was previously defined        System.State.NullifyNegativeValues()                    ▹ CGM can fail to converge if the state contains negative values 
**end for**


**Algorithm 2** Addition of two vectors.
**procedure**Sum(X,λ,Y)                              ▹X←X+λ.Y     **for** each node *i*
**do**          X[i]←X[i]+λ.Y[i]           **end for**
**end procedure**



**Algorithm 3** Naive dot product.
  **procedure**NaiveDot(X,Y)        Z←ElementWiseProduct(X,Y)         **for** each node *i*
**do**              Stride←1              **while**
i%2.Stride==0**and**i+Stride<N
**do**                   Z[i]←Z[i]+Z[i+Stride]                  Stride←2.Stride                  SYNCHRONIZE()          **end while**      **end for**      **return**
Z[0]
**end procedure**



**Algorithm 4** Optimized dot product.
**procedure**Dot(X,Y)    Z←ElementWiseProduct(X,Y)            ▹ Using a static Z avoids allocating memory for every run of the program    GlobalStride←1    **while**
GlobalStride≤N
**do**        BLOCKSUM(Z,GlobalStride)        GlobalStride*=BlockSize▹ In CUDA, block size is capped at 1024        SYNCHRONIZE()    **end while**    **return**
X[0]
**end procedure**
    **procedure**BlockSum(Z,α)    **for** each node *i*
**do**        Stride←α        **while**
Stride<BlockSize*α**and**i%(2.Stride)==0**and**i+Stride<N
**do**           Z[i]←Z[i]+Z[i+Stride]           Stride←2.Stride           BLOCKSYNCHRONIZE()        **end while**    **end for**
**end procedure**



### 2.5. Comparison GPU and CPU

We compared the performances of the GPU and the CPU with the same algorithm (pointwise operation for the chemical reaction operator, and conjugate gradient method for the diffusion operator). We solved the system on GPU with a NVIDIA GPU Titan V (5120 CUDA cores, 12 GB memory, peak performance of 14.90 TFLOPS for FP32 and and 7.450 TFLOPS for FP64). We solved the system on CPU with a Intel Xeon Gold 6148 (20 Cores, 40 threads, base frequency 2.40 Ghz) equipped with 188 GB ECC RAM. We estimated the power consumed by the GPU with the *nvidia-smi* command, and the power consumed by the CPU with the *powerstat* command (the power consumed by other electronic components such as the RAM or the disk is negligible). Simulations were performed in a Linux Mint 19.2 environment, with the CUDA library 10.2 and the NVIDIA driver 440.

### 2.6. Post-Processing

We plotted the time-lapses of the PDEs by drawing a rectangle at the location of each node in the mesh, the color of the rectangle indicating the local level prey and predators.

## 3. Results

### 3.1. Geometry

The simulations agree with experiments performed by other groups in microfluidics devices [39]. The system generates traveling waves of preys, closely followed by a massive front of predators (Figure 2). The waves propagate parallel to the walls of the maze, turn at corners and split at junctions into multiple waves that explore their own branch of the junction. The waves go extinct in cul-de-sacs, which can be explained by the unidirectional propagation of the waves: the preys get “cornered” and cannot escape from the predators.

Our maze represents a challenging testbed for the FEM method, because it exhibits many sharp corners where it suddenly turns at a right angle. The normal vector to the wall is not smooth at these corners, which is known to induce difficulties for the FEM [49]. They are mitigated by finely graining the mesh near the sharp corners. Simulations confirm that the quality of the mesh is crucial to correctly solve the equations (Figure 3). Coarse meshes (∼7 k nodes) do not yield the same dynamic as fine meshes (∼1 M nodes). The prey is found to explore the maze much more quickly on small meshes, which suggests that coarse-graining produces numerical artifacts. We also observe the spontaneous appearance of preys far existing preys, which is not physically possible given the diffusive nature of the system: new chemical species can only appear close to existing ones.

### 3.2. Comparison Performance

We studied the wall-clock time needed to simulate the system on CPU and GPU for 5000 steps (Figure 4). Overall the GPU is faster than the CPU, except for the smallest mesh size where the CPU is slightly faster. The speedup factor of the GPU over the CPU grows from ∼0.9 for the smallest mesh up to ∼130 for the largest mesh, where it plateaus. This speedup is over a multi-threaded implementation on a 20 cores (40 threads) CPU, which represents a substantial improvement over reported speedups for GPUs and FEM [51,57,77]. We attribute this speed up to an optimal use of the architecture of the GPU.

Surprisingly, the wall-clock time for simulation initially decreased with matrix size for the GPU. We found that the conjugate gradient method actually needed less and less iterations to arrive at the required precision as the matrices grew in size. Figure 5A shows for each matrix size, the number of iterations needed for the conjugate gradient method in function of the step number. For smaller matrices, the mean number of iterations is large and varies widely, but as the matrices get larger, the number of iterations becomes tamer and fewer. The mean number of iterations decreases with the matrix size (Figure 5B). We attribute this counter-intuitive behavior to the better conditioning of the matrices, which steadily decrease with finer meshes, almost reaching the minimum value of 1 (Figure 5C). We hypothesize that as the meshes become finer and finer, diffusion becomes more and more regular, locally resembling diffusion on a regular grid. More generally, sharp corners are better approximated by finer meshes, which likely helps in reducing the conditioning number [78,79].

We also profiled the power consumption and usage of GPU/CPU (Figure 4). The power consumption and usage of the GPU grew steadily and monotonously with matrix size. For the largest matrix, the power consumption was close to its theoretical maximum: ∼220 W for a thermal design power of 250 W. The largest matrices also come close to fully utilizing the computational power of the GPU, with an average usage ∼90%. This is remarkable, as it is often challenging to fully mobilize the parallel architecture of GPUs on a single real-world problem [4]. This trend was reverted for the CPU. The CPU usage decreased from 100% to ∼30% as the matrix size increased over ∼50,000 nodes, and the power consumption followed a similar trend. This pattern may be due to a switch from a compute-bound regime (where computation by the CPU is the bottleneck) to a memory-bound regime (where memory accesses to the matrices become the bottleneck). This under-usage of the CPU is in stark contrast with the almost complete usage of the GPU for large matrices, and clearly shows the superiority of the latter over the former for handling large problems in FEM.

### 3.3. Profiling

We profiled the time spent by the algorithm in each portion of the resolution. For the largest matrices, the reaction step only used 2% of the computation time, and 9% of the time was devoted to computing the matrix (D+τS), which is done only once at the beginning of the computation. Solving the linear system for the diffusion step with the conjugate gradient represents ∼90% of the total computation time. This diffusion step itself is broken down into taking the dot product of vectors (57% of the total time), summing vectors (18%) and initializing the vectors of the conjugate gradient (12%). Matrix multiplication only took 1% of the total computation time.

At first, it could seem counter-intuitive that much more time is spent on the dot product of vectors rather than the product of the matrix with vectors (which in itself is a collection of dot products between the rows of the matrix and the vector). However it must be remembered that the matrix is sparse, so that this dot product is done between a sparse vector and a dense vector, which requires only ∼*a* operations (where a<<m is the mean number of non-null elements per row in the matrix, and *m* the length of the multiplied vector). On the other side, taking the dot products of two dense vectors of length *m* is expected to require ∼*m* operations, and it cannot be fully parallelized, because intermediate dot products must be synchronized between the computing threads, which takes at least $log_{2}\left(m\right)+1$ steps [80].

## 4. Discussion

We presented a framework for solving biochemical reaction-diffusion PDEs in arbitrary geometries with FEM on GPUs, which could benefit the DNA nanotechnology community to prototype and debug DNA-based reaction-diffusion systems [39,45,47]. The algorithm fully exploits the massive parallelism of GPUs, achieving a speedup of up to ∼130 over the same algorithm executed on a CPU. We identified a few bottlenecks to improve the future performances. Somewhat ironically, the processing of the PDEs with the GPU is so fast that pre-processing (initial parsing of the matrices) and post-processing (producing the time-lapse movie of the solution) become the bottleneck. Parsing large matrices from the CSV file only represents a few percent of the computation time of the CPU, but exceeds the computation time of the GPU. Storing the matrices directly in a float format would save the lengthy conversion from text to float. Alternatively, it would be beneficial to encode the matrices into a format optimized for vector architectures, such as the NVIDIA PKT format [80]. Moreover, post-processing operations are highly parallel, and in future implementations could be performed directly by the GPU, enabling real-time visualization of the solutions.

Additionally, more than half of the time was spent on computing dot products between dense vectors. Execution could potentially be accelerated with algorithms that use fewer dot products and rely almost exclusively on sparse matrix-vector products. Gradient descent is a candidate for this, though it does not enjoy the same speed of convergence as the conjugate gradient method for symmetric and positive-definite matrix. More simply, some dots products could be saved by reusing them between two successive iterations of the conjugate gradient, and the norm of the residual could be tested every other iteration.

Alternatively, it might be possible to increase the performance of the algorithm by applying different solvers, such as the conjugate gradient squared method or the Chebyshev iteration method [73]. However, these methods usually have an higher complexity per iterations compared to the baseline of the conjugate gradient, so these method would have to significantly reduce the number of iterations to be competitive.

Overall, this work shows that the massive parallelism of GPUs make them a powerful tool to speed up the simulation of PDEs and geometries used in DNA nanotechnology. We limited ourselves to reaction-diffusion PDEs, but similar equations, such as the advection-reaction-diffusion equation, could in principle be tackled (though the mathematical framework would be more involved because advection is a transport operator, which is hyperbolic) (See Supplementary Materials for details).

## Figures and Tables

**Figure 1 micromachines-11-00881-f001:**
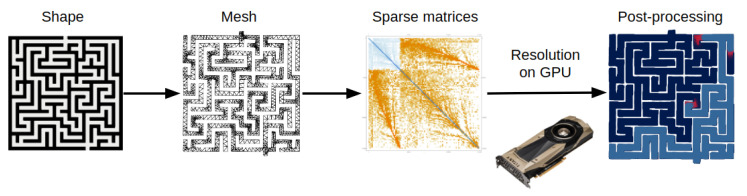
Workflow of our methodology. We start from a bitmap image of a maze that is discretized into a mesh with a FEM software, which then assembles the stiffness and damping matrices. We solve the matrix ordinary differential equations on the GPU, and then plot the results with the CPU.

**Figure 2 micromachines-11-00881-f002:**
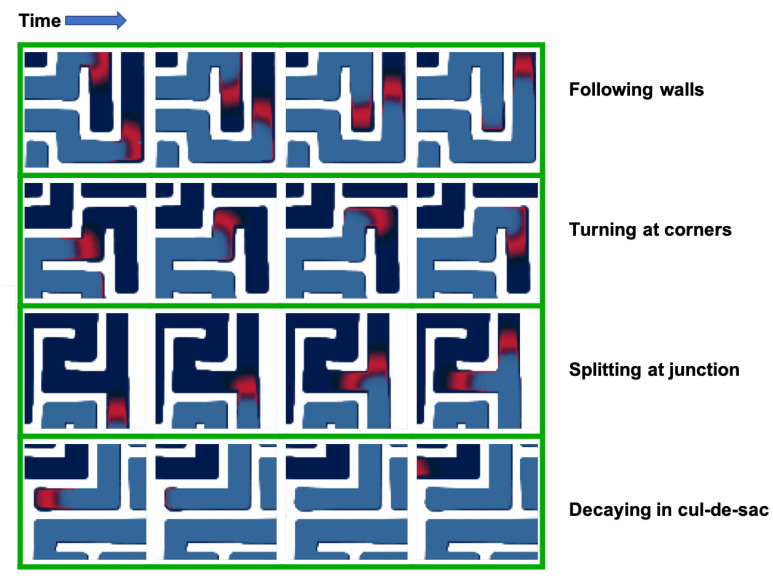
Effect of boundary conditions on the dynamic of the system. The predator-prey system generates traveling waves of preys (in red) followed by a front of predators (blue). The traveling waves closely interact with the geometry of the reactor. They follow straight walls, turn at corners, split at junction, and go extinct in cul-de-sacs. For each geometrical effect, four representative snapshots are presented (taken at an interval of 2000 time-steps).

**Figure 3 micromachines-11-00881-f003:**
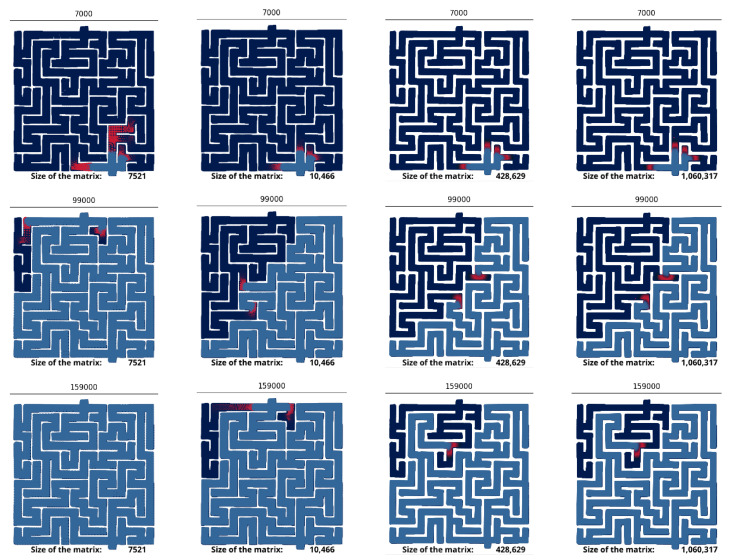
Numerical artifacts for coarse grained meshes. Snapshot of the simulations for coarse and fine meshes. The time-step is indicated at the top of each snapshot. The size of the mesh is indicated at the bottom of the snapshots. The prey is colored in red, and the predator colored in blue.

**Figure 4 micromachines-11-00881-f004:**
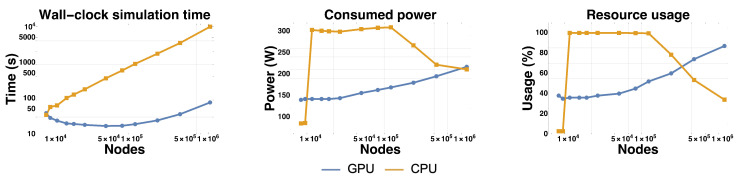
Comparison of performances of the GPU and the CPU for 5000 steps. The plots compare resolution on the GPU and on the CPU for 3 metrics of interests: wall-clock simulation time, average power consumption and resource usage.

**Figure 5 micromachines-11-00881-f005:**
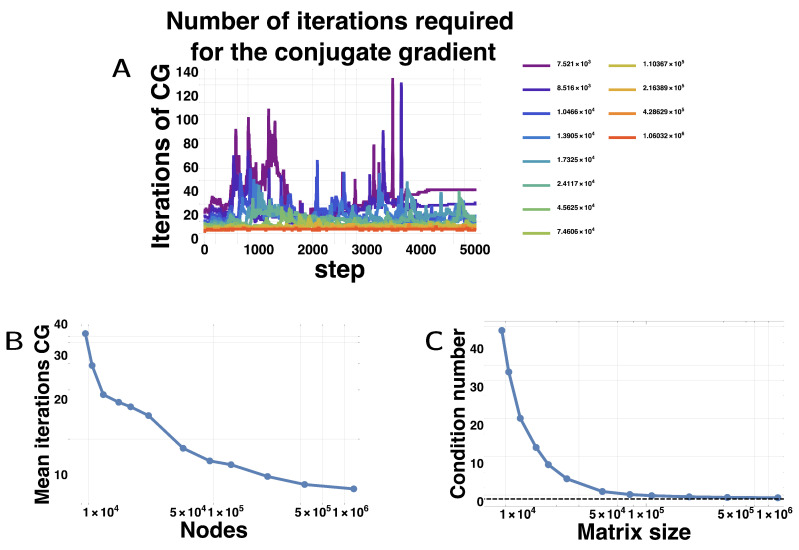
Numerical difficulty of the Conjugate Gradient Method. (**A**) Number of iterations required by the CGM to reach convergence (relative error of the residual decreases below 0.001) for different matrix sizes; (**B**) Mean number of iterations of the CGM extracted from (**A**); (**C**) condition number of the matrix (D+τS) according to the matrix size. The dashed line shows the value of 1, which is the absolute minimum for a condition number. The condition number of the matrix was taken by multiplying the $L^{2}$ norm of the matrix and its inverse, each estimated on random vectors of norm 1.

**Table 1 micromachines-11-00881-t001:** **Examples of representative works on GPU-accelerated resolutions of reaction-diffusion PDEs**. These methods include Finite Difference Methods (FDM) and Finite Element Methods (FEM) [55]. For each study, we present a typical speed-up between CPU and GPU implementations.

Paper	Method	Problem	Speedup (CPU vs. GPU)
Sanderson et al., 2009 [52]	FDM	grid mesh, Advection-Reaction-Diffusion	∼5–10x vs. one CPU core
Molnar et al., 2011 [50]	FDM	grid mesh, Turing Patterns, Cahn–Hilliard eq., …	∼5–40x vs. one CPU thread
Pera et al., 2019 [56]	FDM	grid mesh, tumor growth	∼100–500x vs. 8-core CPU
Gormantara et al., 2020 [57]	FDM	grid mesh, FitzHugh-Nahumo model	∼10x vs. CPU
Sato et al., 2009 [53]	FEM+ODE	3D cardiac simulations	∼0.6x vs. 32-CPU cluster
Mena et al., 2015 [54]	FEM+ODE	3D cardiac simulations	∼50x vs. one CPU core
Descombes et al., 2015 [51]	FEM	chemotactic reaction-diffusion on arbitrary surface	∼100–300x vs. 4-core CPU

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
