# Peer review of "Accelerating the Finite-Element Method for Reaction-Diffusion Simulations on GPUs with CUDA"

_micromachines, 2020, doi:10.3390/mi11090881_

Round 1
Reviewer 1 Report
The paper is devoted to the description of the implementation of the Finite Element Method (FEM) on GPU by CUDA. In general, the material contains interesting findings and the link to github repository.
However, I disagree with the claimed statement, that "This is reflected by the relative paucity of the literature on FEM and GPUs". Even a brief search allows to find dozens of publications related to the subject, especially if you are not limited by CUDA only, but consider OpenCL, for instance. It would be nice to see a comparison of existing methods with the proposed one in this paper.
Also, there are several mistakes in the text and formatting, for example, double "suddenly" in line 167 and figure caption after figure 5.
Reviewer 2 Report
Thank you for the opportunity to review this paper.
The paper is interesting, clearly written. The authors did a lot of work.
But I have a few comments:
- a clear primary goal of the work is not clear from the abstract and from the paper. Is the primary goal of the reaction-diffusion solution (predator-prey system) in zigzag geometry or a demonstration of acceleration of the calculation using the GPU?
- the issue of using GPU in FEM applications has been studied for a long time (eg Ferenc Molnár, Ferenc Izsák, Róbert Mészáros, István Lagzi (2011) Simulation of reaction-diffusion processes in three dimensions using CUDA, Chemometrics and Intelligent Laboratory Systems, Volume 108, Issue 1, 15 August 2011, Pages 76-85, Huthwaite, Peter. (2014) .Accelerated finite element elastodynamic simulations using the GPU. Journal of Computational Physics. 257. 687-707. 10.1016 / j.jcp.2013.10.017. , Zhisong Fu, T.James, LewisRobert, M.KirbyRoss, T.Whitaker (2014), Journal of Computational and Applied Mathematics, Volume 257, February 2014, Pages 195-211 and many more). I lack references to similar works in the references.
The work is well written, interesting, I have no comments on the formal page or the content of the paper.
However, this issue is not new, it is already described in many papers. I encourage authors to rewrite the paper thoroughly, highlighting the differences from the previous papers describing the GPU application. Describe clearly the goals and benefits of the work. I recommend to add more references to papers dealing with GPU applications in reaction modelling.
Round 2
Reviewer 2 Report
My main comment was on the ambiguity of the main goal of the paper and also on the novelty of the topic.
The authors edited the text of the paper, abstract. The aim of the paper is now clearly formulated. It is also possible to distinguish the content of the paper from similarly focused papers.
Thank you to the authors, they did a lot of work.